# Extreme differences between human germline and tumor mutation densities are driven by ancestral human-specific deviations

José María Heredia-Genestar [1], Tomàs Marquès-Bonet [1,2,3,4], David Juan [1✉] & Arcadi Navarro [1,2,3,5✉]

Mutations do not accumulate uniformly across the genome. Human germline and tumor mutation density correlate poorly, and each is associated with different genomic features. Here, we use non-human great ape (NHGA) germlines to determine human germline- and tumor-specific deviations from an ancestral-like great ape genome-wide mutational landscape. Strikingly, we find that the distribution of mutation densities in tumors presents a stronger correlation with NHGA than with human germlines. This effect is driven by human-specific differences in the distribution of mutations at non-CpG sites. We propose that ancestral human demographic events, together with the human-specific mutation slowdown, disrupted the human genome-wide distribution of mutation densities. Tumors partially recover this distribution by accumulating preneoplastic-like somatic mutations. Our results highlight the potential utility of using NHGA population data, rather than human controls, to establish the expected mutational background of healthy somatic cells.

[1] Institute of Evolutionary Biology (CSIC-UPF), Department of Experimental and Health Sciences, Universitat Pompeu Fabra, 08003 Barcelona, Spain. [2] CRG-CNAG, Centre for Genomic Regulation (CRG), Barcelona Institute of Science and Technology (BIST), 08003 Barcelona, Spain. [3] Institució Catalana de Recerca i Estudis Avançats (ICREA), 08010 Barcelona, Spain. [4] Institut Català de Paleontologia Miquel Crusafont, Universitat Autònoma de Barcelona, 08193 Cerdanyola del Vallès, Barcelona, Spain. [5] Barcelonaβeta Brain Research Center (BBRC), Pasqual Maragall Foundation, 08005 Barcelona, Spain. ✉email: david.juan@upf.edu; arcadi.navarro@upf.edu

Mutation density, at different scales, has been shown to correlate with different genomic features, such as regional GC-content or recombination rate[1–5]. In cancer, mutation density shows a different behavior than in human germline and has been linked to chromatin states[6], with higher mutation accumulation in closed chromatin. It has been suggested that the tumor mutation density distribution is mediated by the different replication timing of euchromatin and heterochromatin regions. Closed-chromatin, late-replicating regions have poorer accessibility or recruitment of the mismatch repair machinery, decreasing the efficiency of DNA damage repair in them[7,8]. The correlation between tumor mutation density and chromatin state is tissue-dependent, allowing the identification of the tissue-of-origin of metastatic tumor samples[9,10]. Recent studies have detected mutation patterns in human healthy somatic tissues resembling those seen in tumors of that same tissue[11,12]. These patterns suggest the presence of inherent differences between the mutation landscapes of the human germline and soma.

At a smaller scale, sequence context is a good predictor of the mutation rate[13], even beyond hypermutable CpG sites[14–17]. Sequence context has been widely used in cancer analyses to detect signatures of mutation associated with mutagens such as UV-light, tobacco smoke, or APOBEC activity[18,19]. These effects have also been detected in healthy somatic tissues[11,12]. Interestingly, it has been shown that the human mutation spectrum can be recapitulated by a combination of two signatures: SBS1, associated with CpG>T transitions and SBS5, associated with a slight increase in many non-CpG mutations[20,21]. These two signatures are detected in virtually all cancer samples[19]. The signatures' mutation load is correlated with age of diagnosis and their prevalence in most tumors suggests that many of the cancer mutations occur in preneoplastic stages[19,21].

De novo mutations are also affected by sequence context[20,22,23]. The rates of some particular mutation types have changed recently across ancestries[24–27], and seem to have been under selection in the human lineage[24]. Sequence context studies have shown differences in the relative proportion of certain mutation types between great ape species[26]. Furthermore, studies of de novo mutations in great ape samples revealed a slowdown of the overall mutation rate in humans relative to chimpanzees and gorillas[28].

These previous works suggest that the mutation density landscape in the human germline has been influenced by specific phenomena and that part of the mutations accumulating in tumors might be driven by the same processes that occur in healthy cells. However, little is known about the forces driving mutation accumulation in the human germline and their impact on the pronounced differences in mutation densities between tumors and the human germline. Here, we use NHGA population data to understand the origin of these differences by analyzing their evolution through the human lineage. We analyzed mutation distribution (at the 1 Mbp scale) in human and NHGAs (our closest relatives), and compared them with the distribution of somatic mutations in tumors. This evolutionary analysis allows us to study the conservation status of mutation density, as well as to detect species-specific mutation hotspots and coldspots, thus shedding light on the processes governing differential mutagenesis across the genome.

We observe that the mutation distribution in NHGAs and in tumors present striking similitudes. The differences with human mutation distribution are driven by changes in human's past demography affecting the distribution of CpG>T transitions and mutations at non-CpG sites. We do not detect any mutational process driving the NHGA-tumor similitudes, suggesting that these similitudes are driven by the normal accumulation of mutations in healthy human cells.

## Results

**Mutation density distribution**. We compared the mutation density distribution in human (1KGP[29], SGDP50 (ref. [30])), non-human great apes (NHGA: chimpanzee[31,32], gorilla[31,33]), and human cancer[34] data sets. We focused on high-quality orthologous regions shared between human, chimpanzee and gorilla genomes, measuring the number of variants per 1 Mbp independently of the frequency of each variant (see Methods, Supplementary Note 1).

In agreement with previous reports[1,3,4,6], we observe a variable distribution of the mutation density across the genome in all data sets (Fig. 1a) and a weak correlation of mutation densities of the human germline and tumors[1,6] (Table 1). Strikingly, the NHGA–tumor correlations are much stronger than the human–tumor correlation and are similar to the human–NHGA germline correlations (Table 1 and Supplementary Tables 1, 2).

We compared the distribution of mutation density between pairs of data sets (Supplementary Fig. 1). Interestingly, we observed that mutation density in tumors is higher in windows where NHGAs have higher mutation density than humans (Fig. 1b, c). To control for differences in the shapes of distributions, we ranked each set of windows according to their mutation density (Fig. 1d, e). These ranked distributions show a clear pattern: tumor mutation densities are higher in windows with higher ranks in NHGAs than in human (two-sided Mann–Whitney $U$ test $p$ value human–chimpanzee = 3.7e−216; human–gorilla = 2.8e−161). This human-NHGA-tumor diagonal behavior is exclusive to human-NHGA comparisons, as it cannot be observed when comparing chimpanzee to gorilla ($p$ value chimpanzee–gorilla = 0.28) (Supplementary Fig. 1), and can be replicated under different conditions, in orangutan, and when using other human data sets (Supplementary Notes 2, 3, Supplementary Tables 3–12, and Supplementary Fig. 1).

**Subspecies' diversity**. One possible explanation for this observation is the higher genetic diversity of chimpanzees and gorillas relative to human populations. We explored this by analyzing individual NHGA subspecies with varying levels of diversity. High-diversity NHGA subspecies have stronger correlations with both human and tumor than the low-diversity subspecies (Supplementary Note 3, Supplementary Table 5). In fact, high-diversity subspecies present NHGA-tumor correlations as strong as when analyzing the whole species. This suggests that genetic diversity is one of the drivers of the NHGA-tumor correlations. Furthermore, the diagonal pattern is only characteristic of comparisons between the germlines of humans and high-diversity NHGA subspecies. A comparison of high and low-diversity chimpanzee and gorilla subspecies showed a clear horizontal split (Supplementary Fig. 2). Mutation density in tumors co-localizes with the most diverse NHGA subspecies, regardless of the mutation density in the least diverse. However, the diagonal pattern could never be reconstructed by comparing two NHGA subspecies. In other words; while a lack of diversity distorts the distribution of the genome-wide mutation densities, the diagonal pattern is caused by effects intrinsic to the human lineage.

Consistent with this idea, we observed a weak intermediate pattern when comparing NHGA to three archaic hominid genomes (Supplementary Note 3, Supplementary Fig. 1, Supplementary Table 13). Although the effects of small sample sizes and of ancient DNA damage should not be disregarded, these results suggest that at least part of the differentiation process in the distribution of mutation densities was already established before the human–Neanderthal split.

**Chromatin state and genomic features**. Previous studies have shown that tumor mutation distribution is associated with the

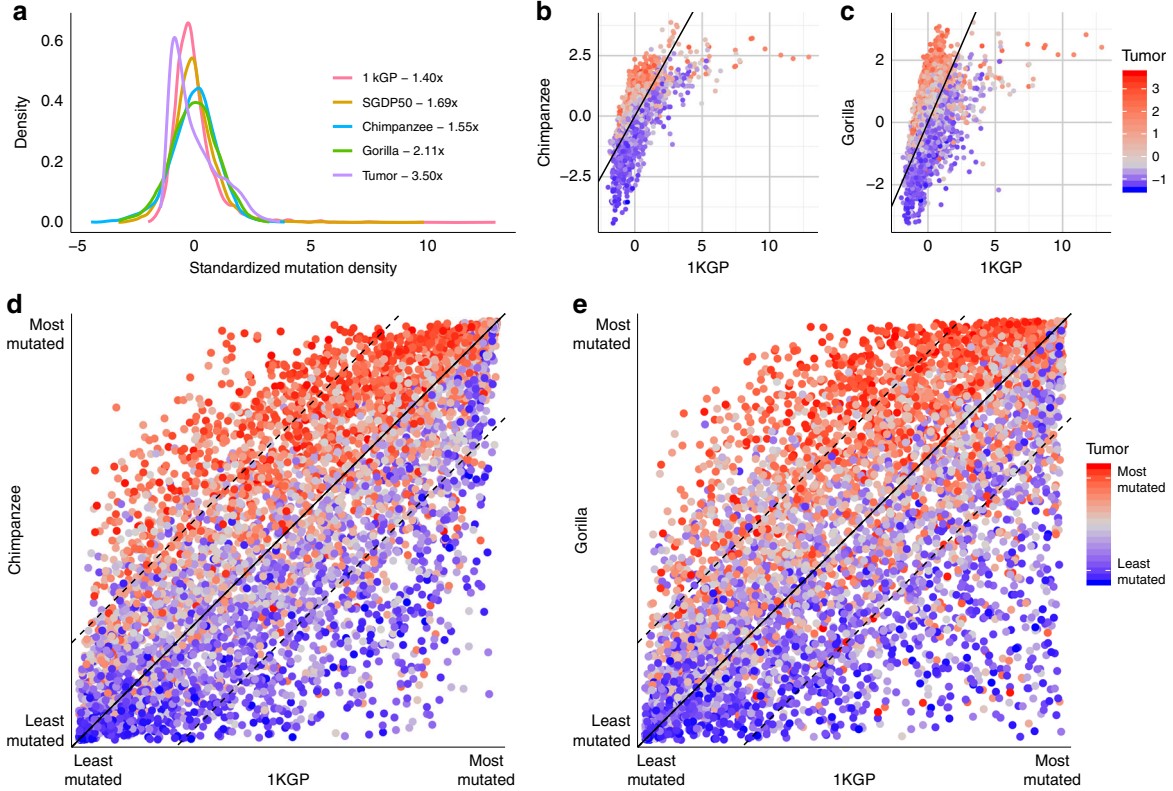

**Fig. 1 Distribution of mutation density across data sets. a** Distribution of the standardized mutation density in 1 Mbp windows in human, NHGA, and tumor data sets. Color denotes data set. The numbers next to the legend are the fold-enrichment between the 95th and 5th quantiles. **b** Distribution of the standardized mutation density in humans, chimpanzee, and tumor. Each point represents a 1 Mbp window. The *x* axis represents the human mutation density, the *y* axis the chimpanzee mutation density, and the point color, the tumor mutation density. The black line corresponds to the diagonal where the mutation density is equal in human and chimpanzee. **c** Same as (**b**) but comparing human and gorilla. **d** Distribution of the ranked mutation density in humans, chimpanzee, and tumor. Each point represents a 1 Mbp window. The *x* and *y* axis represent the ranking in mutation density in human and chimpanzee, respectively. Color represents the ranked mutation density in the tumor data set. The solid black line corresponds to the diagonal where the ranked mutation density is equal in human and chimpanzee. The dashed lines represent a 25% difference in ranking in both species. **e** Same as (**d**), comparing human and gorilla.

**Table 1 Correlation between distributions of mutation density.**

|            | 1KGP | Chimpanzee | Gorilla |
|------------|------|------------|---------|
| Chimpanzee | 0.65 | –          | –       |
| Gorilla    | 0.53 | 0.84       | –       |
| Tumor      | 0.16 | 0.55       | 0.58    |

Pairwise Pearson's correlation *R* of the standardized mutation density of 5040 1 Mbp windows between data sets.

tumor's tissue-of-origin chromatin conformation[9]. When analyzing individual tumor types separately, we detect the human-NHGA-tumor differential pattern across all tumor types. Although the mutation density of tumors across different tissues is highly correlated, some high-sample size tumors (skin melanoma and breast adenocarcinoma) present slightly lower correlations (two-sided Mann–Whitney *U* test *p* value skin melanoma = 1.8e−118; breast adenocarcinoma = 5e−86) (Supplementary Note 3, Supplementary Tables 14, 15).

Interestingly, correlations between a variety of genomic features and tumor mutation density are consistently more similar to the correlations with NHGAs than with humans (Fig. 2a). Mutation densities in NHGAs have, like in humans, strong correlations with sequence conservation and recombination rate (Supplementary

Note 4, Supplementary Fig. 3). However, and strikingly, NHGAs show strong positive correlations with epigenomic features associated with closed chromatin (derived from lymphoblastoid cell lines), just as tumors do, but humans do not (Fig. 2a, Supplementary Table 16). We also observe consistent associations with human chromatin states[35] (Fig. 2b, c). GC-content, CpG-content, and H3K36me1 (specifically recruited in the gene bodies of genes regulated by CpG islands)[36] show a clear positive correlation with human but negative with NHGAs and tumors, suggesting that they might be contributing to the diagonal pattern (Figs. 2d, e, 3a, b).

**CpG sites.** Intrigued by the connection of several CpG-related features with the human-NHGA-tumor diagonal pattern that implies a stronger correlation between mutation densities in tumors and NHGAs than with the human germline (Fig. 3a, b), we analyzed separately CpG>T transitions and mutations at non-CpG sites. CpG>T transitions present very strong correlations between all germline data sets and very poor correlations with tumor, driven by their strong association with CpG-content (Fig. 3c, d). The relationship between CpG-content and mutation density at non-CpG sites in NHGAs is more similar to tumors than to the human germline. Moreover, the correlations of non-CpG mutation densities between human, NHGA, and tumors are similar to those observed using all sites (Fig. 3e, f, Supplementary Tables 17–20). The correlation of genomic features with CpG>T transitions are similar between germline data sets, while the

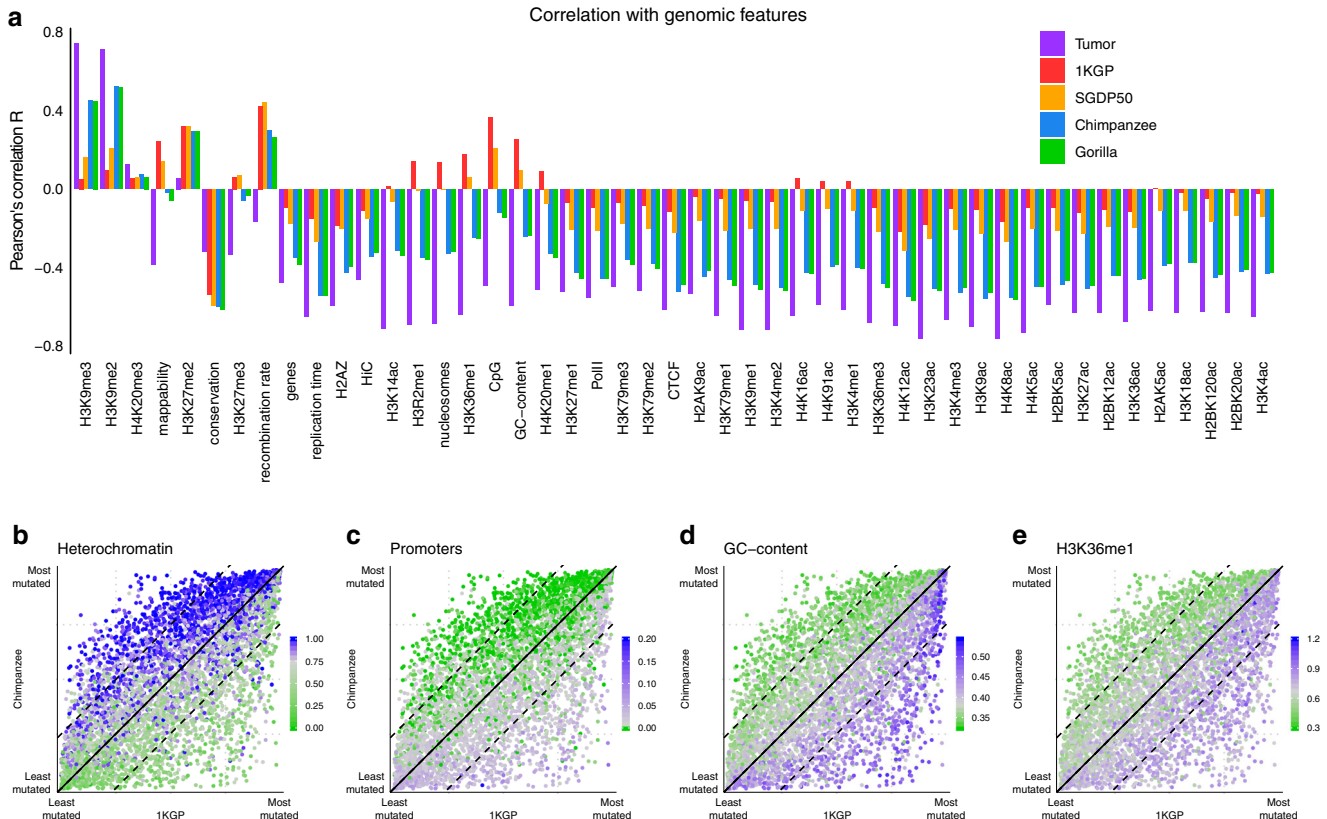

**Fig. 2 Genomic Features. a** Pearson's correlation *R* of different data sets with human genomic features (data sources in Supplementary Table 16). Color denotes data set. **b** Human and chimpanzee ranked mutation density distribution. Color represents the overlap of heterochromatin in human lymphoblastoid cell lines (LCLs) measured by chromHMM states[35], **c** the aggregate chromHMM states associated with the presence of promoters, **d** the window's GC-content, and **e** the density of H3K36me1 histone mark ChIP-seq reads[57].

correlations of non-CpG sites are weaker in humans than in NHGA and tumors (Supplementary Note 5, Supplementary Fig. 2). Consequently, while tumor-specific differences in correlations of mutation densities with genomic features are associated with CpG>T mutations, human germline-specific differences are related to the different behavior of human non-CpG mutations.

Despite the overall mutation slowdown previously described in human[28,37], all data sets present similar proportions of recurrent CpG>T transitions, discarding a possible effect of CpG>T saturation on these human-exclusive differences (Supplementary Note 5, Supplementary Tables 21 and 22). Interestingly, we observed that even a small number of de novo mutations[38] can recover much higher correlations with all germline data sets for CpG>T than for non-CpG mutation densities (Supplementary Table 23). This observation suggests that CpG>T mutations require much less diversity than non-CpG mutations to recover its genome-wide distribution, probably due to their strong dependence of the skewed genome-wide distribution of CpG-content. In contrast, the distribution of human non-CpG de novo mutations replicates the general trend of all germline data sets showing intermediate but higher correlation with NHGAs than with tumors (Supplementary Note 5, Supplementary Table 23), however, the small size of this data set makes difficult to discern whether their correlation with tumors is lower than expected for NHGAs.

When comparing the distribution of non-CpG mutations in human, NHGA, and tumors, we detect a horizontal pattern (Supplementary Fig. 3) similar to those observed in comparisons of high- and low-diversity subspecies. This suggests that the effects of loss of diversity in a given species can be detected mostly in non-CpG sites but not in CpG>T mutations. In fact, within-

species correlation of CpG>T transitions and mutations at non-CpG sites is weaker in humans than in NHGAs and in tumors (Supplementary Fig. 3, Supplementary Table 24). Therefore, the combination of human-specific dynamics in both, non-CpG and CpG>T mutations, causes the diagonal pattern observed when comparing all SNVs.

**Mutational signatures.** Studies of cancer mutations have associated certain mutational signatures with the action of specific mutagens or cellular processes[18]. We benefited from these associations to explore the contribution of different mutation mechanisms to the observed differences in mutation densities by analyzing the trinucleotide context of SNV in all germline data sets.

At the whole-genome level, the triplet mutation spectra of human, chimpanzee, and gorilla are very similar (Supplementary Fig. 4, Supplementary Table 25), although different from tumors. It has been shown that the human mutation spectrum can be recapitulated by a combination of the cancer signatures SBS1 and SBS5 (refs. [20,21]). We were able to replicate this association in NHGA and in another primate species (Vervet monkey) (Supplementary Note 6, Supplementary Tables 26 and 27). Previous studies have shown that both, germline and somatic, mutation spectra of mice differ from the human mutation spectra[39]. Our results show that mutation spectra and their specific association with signatures SBS1 and SBS5 are conserved across the primate lineage.

A subset of trinucleotides show significant species-specific enrichments (Chi-Squared test *p* value < $10^{-5}$). We detected no association between these trinucleotides and known mutation mechanisms (Fig. 4a, see Methods, Supplementary Note 6,

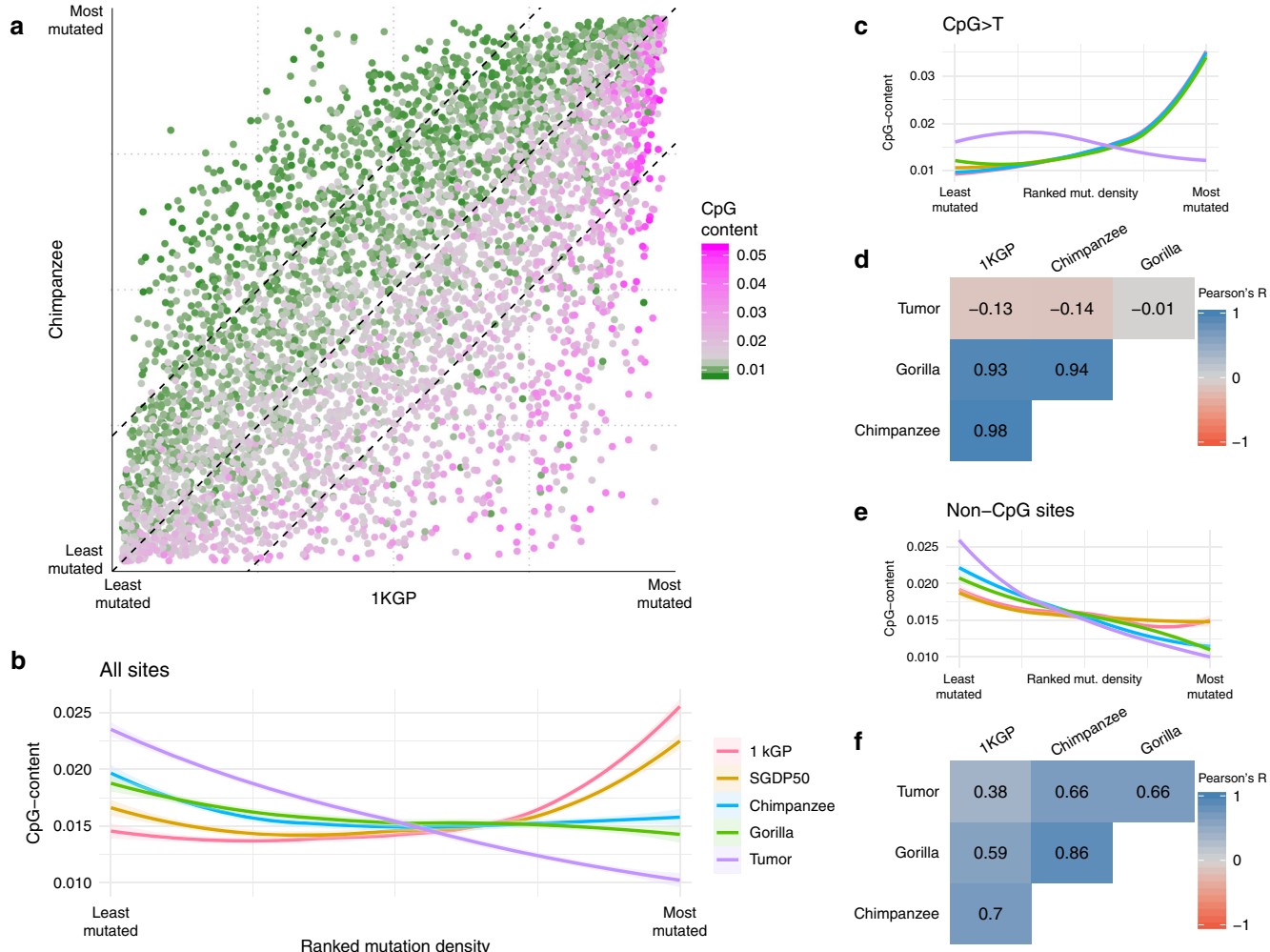

**Fig. 3 CpG-content. a** Ranked mutation density in human and chimpanzee. Color represents the distribution of CpG-content (fraction of the window overlapping CpG sites) in the human reference hg19. **b** Loess smoothers of ranked mutation density and CpG-content for the different data sets. Color denotes data set. Shaded area represents the 95% CI. **c** CpG>T transitions corrected by the whole window size; loess smoothers same as in (**b**); **d** Pearson's correlation R of the standardized mutation density of CpG>T transitions in different species; **e** same as in (**b**, **c**), but using only mutations at non-CpG sites; **f** Pearson's correlation R of the standardized mutation density of mutations at non-CpG sites in different species.

Supplementary Fig. 4, Supplementary Table 28). However, linear regression models show a positive and significant ($p$ value $< 10^{-4}$) effect of the triplet's GC-content and its fold-enrichment in the human–chimpanzee comparison (Supplementary Fig. 4).

Although no specific known mutational mechanism could be detected having a whole-genome effect in the mutation spectrum, other processes might have an effect on the mutation distribution. To test this, we analyzed the differential distribution of trinucleotide mutations in 1 Mbp windows across the genome in humans, NHGA, and tumors. Only trinucleotides with similar enrichment between species (non-CpG, mainly C>G and T>C) show differences in their distribution across the genome between human, NHGA, and tumor (trinucleotide-difference test $p$ value $< 10^{-5}$, see Methods, Supplementary Note 6, Fig. 4a).

We further analyzed this trinucleotide distribution across the genome in individual tumor types. We compared the association of the number of mutations caused by each cancer signature[19] in each individual tumor type to the human-NHGA-tumor pattern (Supplementary Data 1). Signatures SBS5 and SBS40 show a significant association (signature-difference test $p$ value $< 10^{-4}$, see Methods) of the pattern with the tumor's signature mutation load (Fig. 4b). Both SBS5 and SBS40 are flat signatures whose mutation load is associated with the age of diagnosis of the

sample and with preneoplastic mutations in tumors[19,40]. This suggests that the strong correlation between NHGA and tumor mutation densities is driven by the same mechanisms in healthy human cells and in the great ape lineage, while the genome-wide distribution of mutations has been altered in the human germline.

## Discussion

We analyzed the mutation density distribution at the 1 Mbp scale in the human and NHGA germlines, as well as in human tumors. We observed a moderate similitude between human and NHGA germlines, and surprisingly a higher resemblance between human tumors with the germlines of NHGAs than with those of humans.

These discrepancies in mutation density in the human and NHGA germlines are differently associated with genomic and epigenomic features. In fact, human germlines show a lower correlation than NHGA germlines and tumors with most of these features, suggesting that they present a distorted mutational distribution. Regions more densely mutated in humans than in NHGAs tend to be GC-rich, exon-rich, promoter and enhancer-rich, open chromatin and early replicating. Particularly, CpG-related features show a positive correlation with human and a negative correlation with NHGA and tumor mutation densities.

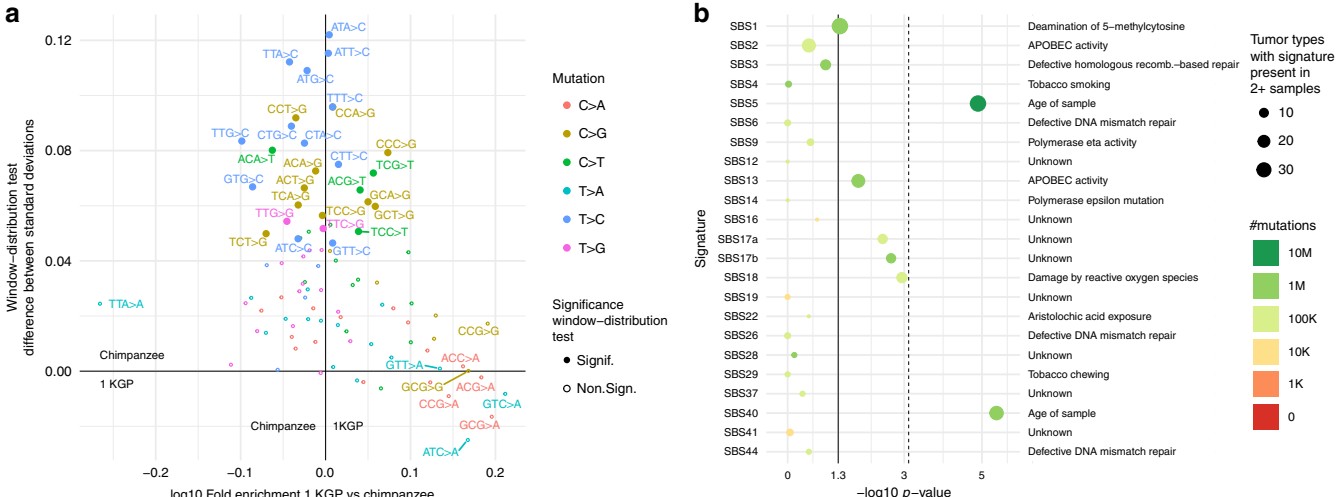

**Fig. 4 Trinucleotide analysis. a** Contribution to the higher chimpanzee-tumor mutation distribution similarity vs. genome-wide enrichment in human compared with chimpanzee. x axis: log10 of the enrichment of trinucleotides comparing human and chimpanzee. Left: enriched in chimpanzee; right: enriched in human. y axis: effect size (difference between the standard deviations of human–tumor and chimpanzee–tumor) of the trinucleotide-difference test (see Methods). Positive values: tumor distribution more similar to chimpanzee; negative values: tumor distribution more similar to human. Color represents the central nucleotide mutation type. Filled dots denote mutation types significant (p value < $10^{-5}$) in the trinucleotide-difference test (Kolmogorov–Smirnov test; multiple testing significance threshold <5 × $10^{-4}$). **b** Chi-squared -log10 p values of the association of each cancer signature mutation load to the trinucleotide-difference test (signature-difference test; see Methods). The dashed line represents the Bonferroni multiple correction significance threshold. Color represents the number of mutations associated with each signature in the whole data set. Dot size represents the number of tumor types with two or more samples showing the signature. Only non-artifact mutation signatures present in two or more tumor types are shown.

The possible functional implications in human evolution require further study.

These observations are driven by the different dynamics of mutation density at CpG>T transitions (very similar in all germlines and very different in tumors) and at non-CpG sites (more similar in NHGAs and human tumors than in the human germline). This effect is exclusive of the human germline and, thus, must have been caused by human-specific conditions.

We observed that human and other primates present a very similar global triplet mutation spectrum. We detected an enrichment of certain trinucleotide mutations in humans and NHGAs consistent with previous results (non-CpG, GC-rich mutations are enriched in humans)[26]. The enriched trinucleotides are not associated with mutation signatures with known causes, nor do they contribute significantly to the higher similitude of human tumors to NHGA germlines. This suggests the absence of strong mechanistic changes biasing the accumulation of mutations in any of the studied germlines.

As previously described for human[20,21], we observed that mutation rates of three non-human primates are explained by mutation signatures SBS1 (mostly CpG>T transitions) and SBS5 (associated with normal accumulation of mutation in healthy somatic and germline cells)[18,20,21,41]. Moreover, the lower human-tumor than NHGA-tumor mutation distribution correlation is driven by the accumulation of mutations associated with signatures SBS5 and SBS40 (similar to SBS5 and recently discovered)[19]. These results suggest that the poor human-tumor correlation is caused by the fact that human (but not NHGAs) germline (and de novo mutations) do not currently reflect the expected mutation densities of healthy (and preneoplastic-like) human somatic cells.

We observed that the moderate human-NHGA and the low human-tumor correlations of mutation densities at non-CpG sites could be caused by losses in population diversity (as observed in low-diversity NHGA subspecies). We propose that successive bottlenecks during human evolution removed a substantial part of nucleotide variation that still remains to be

recovered as a whole. In contrast, the hypermutability of CpG sites and its concentration in specific regions caused CpG>T transitions to have already recovered diversity levels similar to those of high-diversity NHGAs. However, low-diversity NHGA subspecies show lower CpG>T and higher non-CpG degrees of recovery of their mutation densities than the human germline. This suggests that some human-specific phenomena are influencing the different behavior of human mutational landscapes. We pose that this human-specific behavior can be the result of the combination of at least two very different contributions.

First, the recent human-exclusive population expansions[31,42] are expected to cause an increase of clock-like CpG>T mutations in the population[37,43], leaving signatures akin to positive selection, as it has been described in Native Americans[25]. The results presented in our study suggest that these signatures of apparent positive selection may not be population exclusive and might be pervasive through human evolution, especially in CpG-rich exons. Moreover, the highly skewed genome-wide distribution of CpG-content can contribute to the easy recuperation of mutation densities of the fast mutating CpG>T transitions (as observed in a reduced data set of de novo mutations)[38]. The decoupling of the CpG>T/non-CpG mutation rates within the same region is stronger in humans than in NHGAs and tumors and could imply that these population expansions introduce additional distortions on the non-CpG densities associated to specific biases inherited from previously contracted populations. We cannot disregard an additional contribution of human-specific shifts in CpG>T transitions mutation rates, although they have been suggested to be similar across all great apes[37].

Second, the recent slowdown in mutation rates reported in humans[28,37] seems to have affected differently CpG>T and non-CpG sites. A relative deceleration of non-CpG mutations could also contribute to the lesser capability of non-CpG mutations to recover their expected genome-wide distribution. In addition, it could contribute to the CpG>T/non-CpG differentiation, which is observed more strongly in the human germline. Interestingly, our analyses with a data set of de novo mutations[38] show an increase

in the contribution of CpG>T mutations compared with all the other data sets. This suggests that the recent slowdown in humans might have been less pronounced in CpG>T mutations. However, the small size of the de novo data sets makes it difficult to interpret the relation of their genome-wide distributions with the other data sets. Further studies of de novo mutations in both humans and NHGAs are needed to elucidate possible differences in the evolution of mutation rates. We propose that the combination of population bottlenecks and expansions, together with the specific nature of the different mutation types, drives the differences observed in the distributions of human mutation densities. Future additional work will be necessary to establish the relevance of these and other phenomena in the observed human-specific behavior of the genome-wide distribution of mutation densities.

Our results imply that accumulated mutations in human populations are a poor proxy of the expected mutational background in healthy somatic cells. In fact, contrary to the common assumption that extreme differences in the mutational landscape of human germline and tumors are due to the abnormal behavior of tumors, our results show that they are driven by human-specific particularities in the accumulation of mutations in the germline. In fact, accumulated mutations in NHGAs (at least at non-CpG sites) happen to be more informative about the normal occurrence of mutations in healthy somatic cells than those accumulated by humans.

Our results do not affect the well-established association between the mutation's observed population frequencies and their relative functional impact[44]. However, they suggest that higher precision in the definition of damaging mutations could be achieved by incorporating data from less biased populations. Interestingly, a recent work exploring missense mutations in population data from NHGAs for the prediction of the impact of human mutations has already given a first major step in that direction[45].

These results have also paramount implications for the understanding of tumors as often normally mutating primate cell populations. This implies that the definition of positively selected shifts in regional mutational burdens could be much refined by correcting mutation expectations by making use of primate populations or even of tumoral cell populations. Moreover, disentangling the contribution of normal mutation processes in tumors (either if they correspond to preneoplastic cells or not) opens the door to establishing them as models to understand somatic mutations in healthy cells, with particular interest in the experimental research of aging and somatic mosaicism.

## Methods

**Data sets used**. For the human data sets we used the release variant calling of 2504 humans from the 1000 Genomes Project[29] (1KGP), our own calling of 50 additional human samples from the Simons Genome Diversity Panel[30] (SGDP50), and de novo mutations from 1548 trios[38] that were mapped to the human reference hg19 using the liftOver tool[46]. We used our own mapping and calling of 69 chimpanzees and bonobos (59 chimpanzees and 10 bonobos, referred to as chimpanzees in short)[31,32] and 43 gorillas[31,33]. We used the release variant calling of three archaic samples: Altai and Vindija 33.19 Neanderthals[47,48], and Denisova[49]. Finally, for the tumor data set, we used the release variant calling of 2583 human tumors from the Pan-Cancer Analysis of Whole Genomes Consortium[34].

**Definition of high-quality orthologous regions**. We mapped and called chimpanzee and bonobo, gorilla, and human (SGDP50) samples to the human reference hg19 using BWA MEM[50] and GATK[51] following the best practices protocols[52,53]. In addition, we removed variants with ≥20% missings, any variant (SNV or InDel) within 5 bp of an InDel (suggesting potential misalignments), and variants where heterozygous samples represented more than 80% of the calls (suggesting potential mismappings or duplicated regions).

To avoid mismappings to the human reference and erroneous estimates of mutation density in the NHGA samples (too low density caused by lack of mapping reads or deletions or too high density caused by collapsed duplications)[54]

we filtered out any region of the human reference genome hg19 failing one of the following criteria: poor mappability of the human reference split into 35 bp k-mers, poor callability in ≥25% of the chimpanzee or gorilla samples, or, matching a known Copy Number Variable region in NHGA samples[55] (Supplementary Note 1). 2052 Mbp of autosomal sequence passed this filtering (76.54% of the non-N base pairs in the human reference autosomes). We divided the autosomes into 1 Mbp overlapping (500 kb) windows and kept all windows where ≥50% of its bases passed our filtering. This left 5040 1 Mbp windows to analyze (Supplementary Fig. 1, Supplementary Table 3).

These filters were applied to all data sets used, including both our callings and external data sets used as released. All SNV counts, trinucleotide counts, and genomic features measurements through this study used only regions passing this filtering. For the analysis of archaic samples, we combined this filtering with the intersection of the callability mask of all three archaic samples. This specific filtering was applied to all data sets when compared with the archaic samples.

**Mutation density**. We used BASH scripts to measure mutation density of each window in each data set by counting either the number of non-fixed segregating sites (in the human, chimpanzee, and gorilla data sets) or the number of somatic mutations (in the tumor and human de novo data sets, accounting repeated mutations as independent mutational events). We divided this count of single nucleotide variants (SNV) by the fraction of the window passing our filtering. This results in a measure of mutations per megabase pair (Mbp) of sequence for each window. We standardized the resulting distribution within each data set deeming it as the mutation density. We ranked all windows within a data set by their distribution of mutation density to control for the different shapes of the data sets distributions.

**Correlations between distributions**. All correlations used in this analysis are Pearson's correlation coefficient (using the R function cor.test) between the standardized mutation densities (unranked) of the two data sets unless otherwise specified. Partial correlations, when used, were calculated using the pcor function from the ppcor R package.

**Significance of the diagonal split**. To measure the significance of the diagonal split pattern observed when comparing the human and NHGA data sets, we divided all windows into two groups depending on if the ranked mutation density is higher in human than NHGAs or vice-versa. We calculated the two-sided Mann–Whitney $U$ test on the variable of interest (usually, the tumor mutation density) on both groups using the R function wilcox.test.

**Genomic features**. The genomic features used were filtered using the same mappability, callability, and copy-number filters used for the mutation density data. The features used were either the overlap of the feature's genomic coordinates with the fraction of the 1 Mbp window passing our filtering (e.g., GC-content, CpG-content) or the average value or intensity of the feature in the passing fraction of the window (e.g., histone marks), depending on the original data (Supplementary Table 16).

**Trinucleotides**. We classified each SNV into the 96 possible combinations of trinucleotides (12 different mutation types, by 16 combinations of the adjacent nucleotides, divided by two when folding them). We determined the adjacent reference sequence of each SNV using the getfasta option of bedtools[56]. We filtered out any variant where the liftOver tool[46] could not map them to the chimpanzee panTro5 or the gorilla gorGor5 reference genomes, or the trinucleotide sequence differed in one of the three reference genomes (accounting for strand). This filter was applied to all windows and we used for our analysis only windows where ≥50% of it passed both the original high-quality orthologous regions filter and this three-reference filter, leaving 4920 windows to use. We applied additional filters requiring the trinucleotide to be species exclusive and to not overlap variants in other species, the borders of our orthologous regions filter, multinucleotide variants, InDels within the same species, or InDels with frequency ≥50% in another species. This resulted in a high-confidence set of species-exclusive trinucleotides where the ancestral and derived alleles could be reliably inferred. This filtering affected more CpG>T than non-CpG sites, due to the recurrent nature of CpG>T transitions (Supplementary Note 6, Supplementary Table 25).

**Mutation spectra**. We calculated each species' mutation spectra as the fraction of all trinucleotides in a data set belonging to one of the 96 trinucleotides. We calculated correlations between data sets using Pearson's correlation (cor.test function in R). We measured the correlation of the mutation spectrum of each species and the combined effect of cancer mutation signatures SBS1 and SBS5 (refs. [19,40]) by the formula: $0.1 \times SBS1 + 0.9 \times SBS5$, as CpG>T transitions are the main components of signature SBS1 and they represent ~10% of the trinucleotides in both the human and NHGA data sets.

**Whole-genome enrichment of trinucleotides**. We calculated the enrichment of trinucleotides and its significance in each germline data set pair (human–chimpanzee, human–gorilla, and chimpanzee–gorilla). We calculated the enrichment of

trinucleotide $T$ between species $A$ and $B$ by dividing *fraction of $T$ in species $A$/fraction of $T$ in species $B$*. We calculated a chi-squared test using a contingency table with the trinucleotide count in species $A$, in species $B$ and the count of the rest of trinucleotides in species $A$, and in species $B$. As the counts of trinucleotides are not independent of each other, we sorted all trinucleotides from most to least significant, and rerun the test by decreasing significance order, while removing the previously used trinucleotides from the count of total trinucleotides[26].

CpG>T transitions are highly affected by the sample size of the data sets. We ran all the tests using both 1KGP and SGDP50 as the human data set. We detected incoherences on the significance and direction of the results in two CpG>T trinucleotides. We report the results using 1KGP where tests using both 1KGP and SGDP50 are coherent in both significance and direction of the enrichment.

The top 10% most enriched trinucleotides in each species pairwise comparison were compared with cancer mutation signatures[40] and reported when the trinucleotide represented ≥5% of the mutations within a signature.

**Trinucleotide-difference test**. We developed a method to determine which trinucleotides contribute significantly to the difference between NHGAs-tumors and human-tumors mutation density correlations.

For each trinucleotide $T$ and each pair of species (human–chimpanzee, human–gorilla, and chimpanzee–gorilla), we subtract the ranked mutation density of $T$ in species $A$ minus the ranking in tumor, and in species $B$ minus tumor. We calculate the two-sided Kolmogorov–Smirnov test (using the $R$ function ks.test) of the two resulting distributions. We use the $p$ value of the ks test as the significance of the test and the difference between the standard deviation of both distributions (as both have a mean of 0) as the test's effect size. The results when using 1KGP or SGDP50 as the human data sets are concordant in the direction of the association, but we discarded the SGDP50 results because the smaller number of SNVs (and of each trinucleotide type) results in lower power when using SGDP50.

**Association of GC-content in the trinucleotide sequence**. We counted the number of cytosine and guanine bases in each trinucleotide and built a linear regression (using the $R$ function glm). The GC-content of the triplet acted as a predictor of the result of the test (the log10 fold-enrichment in the whole-genome enrichment analysis or the difference between the standard deviation of both distributions in the trinucleotide-difference test).

**Signature-difference test**. In order to determine the contribution of each mutation signature to the difference between NHGAs-tumors and human-tumors mutation density correlations, we rerun the trinucleotide-difference test using the 1KGP and chimpanzee data sets, while using the different individual tumor types (Supplementary Data 1). For each trinucleotide, tumor type and mutation signature, we built a linear regression (using $R$'s glm function) where the mutation load of that signature in that tumor type[19] predicted the effect size in the trinucleotide-difference test for that tumor type (Supplementary Note 6). For each signature, we built a contingency table where all 96 trinucleotides where classified by whether being significant or not ($p$ value < 0.05) in the trinucleotide-difference test and the significance of the mutation load in the linear regression model. We ran a chi-squared test on that contingency table and obtained its significance.

**Reporting summary**. Further information on research design is available in the Nature Research Reporting Summary linked to this article.

## Data availability

All the analyses were performed using publicly available data obtained from their original publications. Human data sets: 1000 Genomes Project (ftp://ftp.1000genomes.ebi.ac.uk/vol1/ftp/release/20130502/), Simons Genome Diversity Project (EBI European Nucleotide Archive accession numbers PRJEB9586 [https://www.ebi.ac.uk/ena/data/view/PRJEB9586] and ERP010710 [https://www.ebi.ac.uk/ena/data/view/PRJEB9586]), International Cancer Genome Consortium (https://dcc.icgc.org/releases/PCAWG/germline_variations), and Human de novo mutations (European Variant Archive accession number PRJEB15197 [https://www.ebi.ac.uk/ena/data/view/PRJEB15197]). Non-human great ape data sets: the Great Apes Genome Project (Sequence Read Archive (SRA) accession numbers PRJNA189439 [https://www.ncbi.nlm.nih.gov/bioproject/?term=PRJNA189439] and SRP018689 [https://www.ncbi.nlm.nih.gov/sra/?term=SRP018689]), European Nucleotide Archive (ENA) accession numbers PRJEB15086 [https://www.ncbi.nlm.nih.gov/bioproject/?term=PRJEB15086] (Chimpanzee), PRJEB3220 [https://www.ebi.ac.uk/ena/data/view/PRJEB3220] (Gorilla), and PRJEB19688 [https://www.ncbi.nlm.nih.gov/bioproject/?term=PRJEB19688] (Orangutan). Tumor data sets: the Pan-Cancer Analysis of Whole Genomes (https://dcc.icgc.org/pcawg/). Archaic samples (http://cdna.eva.mpg.de/neandertal/Vindija/). The remaining data are available within the Article, Supplementary Information or available from the authors upon reasonable request.

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

## Acknowledgements

We thank B. Lehner, D. Weghorn, and I. Lobón for their insights discussing the analyses, and C. Warembourg for her statistical analysis help. J.M.H.G is supported by MDM-2014-0370. T.M.B. is supported by BFU2017-86471-P (MINECO/FEDER, UE), Howard Hughes International Early Career and Obra Social "La Caixa". D.J. is supported by Juan de la Cierva fellowship (FJCI-2016-29558) from MICINN. A.N. is supported by AEI-PGC2018-101927-BI00(FEDER/UE), MINECO-BFU2015-68649-P (MINECO/FEDER, UE), the Spanish National Institute of Bioinformatics of the Instituto de Salud Carlos III (PT17/0009/0020), and by FEDER (Fondo Europeo de Desarrollo Regional)/FSE (Fondo Social Europeo). This work was supported by Unidad de Excelencia María de Maexto, funded by the AEI (CEXS2028-000792-M) and Secretaria d'Universitats i Recerca and CERCA Programme del Departament d'Economia i Coneixement de la Generalitat de Catalunya (GRC 2017 SGR 880).

## Author contributions

J.M.H.G. performed all the analyses. J.M.H.G and D.J wrote the paper. T.M.B., D.J., and A.N. conceived and supervised this work. All the authors read and approved the final paper.

## Competing interests

The authors declare no competing interests.
