## [Peer Review File · Nature Communications]

Reviewers' comments:

Reviewer #1 (Remarks to the Author): Expertise in comparative genomics

In the manuscript, "Mutation distribution density in tumors reconstructs human's lost diversity", the authors Heredia-Genestar et al. map the distribution of mutational density in: 1) ~4,000 humans (1000 Genomes project + Simons Diversity Panel); 2) Non-human great apes (69 chimpanzees/bonobos + 43 gorillas); 3) three archaic humans and 4) ~2,500 human tumors (Pan-Cancer Analysis). They then correlate these mutational densities against each other, and find that non-human great apes have a significantly higher correlations with human tumors than the human samples tested – including the archaic samples. The authors then test the distribution of non CpG vs CpG>T and find the density of non CpG in NHGA and human tumors to be more correlated than humans. Additionally, they confirm that germline mutations rates in humans have undergone a recent slowdown. Authors suggest that humans do not have the expected mutation densities may be due to losses in population diversity from bottlenecks and expansions.

Overall, I find this to be a very careful and overall thorough study, with substantial additional testing in the supplement, including various sizes of sliding window for analyses, additional datasets, subsets of humans, subsets of tumors, and variant frequency - suggesting the signal the authors are finding is indeed true. I also find the methods and results to be solid and an important contribution to the field of human evolution and cancer biology. I have a few concerns and questions about the interpretation and potential additional analyses outlined below.

Major/Minor Concerns:

1. The underlying theory, motivation and hypothesis for testing these distributions is a bit unclear in the introduction.

2. I have a hard time with the interpretation of the data. While I believe that the results are true, it's unclear the mechanism of how these tumor mutation distributions are correlated more closely to NHGA than humans. I wonder if the results are highlighting different levels of DNA repair more than the actual mutation burden. It may be that human tumors and NHGA have similar mechanisms of repair. Additionally, tumors have accelerated mutations and distributions, so I wonder if this relationship would hold with all species with faster germline mutation rates.

3. I have a few questions about the samples/data.

a. What were the male/female distribution in the samples – both human, NHGA and tumors?

b. Were any of the subsets age-matched to see if the pattern holds? Mutation signatures likely correlate with age, and likely many of the tumors represent an older population of individuals. How do ages reflect in the human and NHGA data?

4. For the tumor subset analyses – would the authors expect cancer from germline mutations to have a better correlation between human mutation distribution than NHGA and somatic mutation tumors?

5. NHGA have higher SNVs due to between subspecies differences – could this be one possible explanation for the correlation between cancer & NHGA?

6. In the supplement, authors mention regions with low homology between humans and NHGA would show extremely low mutation density in chimpanzee and gorilla because we would not be able to call variants there. I am wondering if this could bias the results and influence how and where cancer mutations land in the genome.

7. Lastly, the conservation of mutation spectra in additional species, including vervet monkey – I was confused on whether the tumor samples showed stronger correlation with human tumors than humans?

Reviewer #2 (Remarks to the Author): Expertise in comparative genomics (cancer)

I found this to be a creative manuscript with novel analyses and enjoyed reading it. The main observation is quite striking, that there is a stronger correlation in the mutational rate across the genome between human tumours and NHGA germline than there is between human tumours and the human germline. This provides new insights into differences between somatic and germline mutations and about changes in mutational rates and patterns across the great ape lineages. The authors speculate about the causes for these observations, with the most likely explanation being past changes in human population size resulting in a decrease in diversity followed by a relative increase in the accumulation of more clock-like CpG > T transitions. I believe this observation is a useful contribution to the field of genomics and will be of interest to those working in the fields of both somatic and germline evolution.

I have two main comments. The first is that the flow of the results section of the manuscript is sometimes difficult to follow and it is not always clear why certain analyses were done. I think adding a few additional sentences before some analyses explaining the motivations could help guide the reader. Having said that the discussion section does a good job of summarising the key findings.

The second main comment is that currently the manuscript presents a striking observation and speculates on the causes without doing any further analyses or experiments to explore them. I don't think this is the fault of the authors as it would probably be very difficult to think of ways to test which explanation for the observations is correct. However if it were possible to do any experiments such as simulation studies of human population size changes and explore whether these do indeed recapitulate the patterns they observe relative to great apes, that would certainly strengthen the paper. However I recognise that this would add substantial work and could probably in itself be a separate manuscript.

Below are some minor comments.

Main Text:

Line 2: Technically I think the word 'human's' would need to be changed to 'humanity's' or 'human-kinds' to be grammatically correct. Also the mutation distribution density in tumors may not reconstruct humanity's lost diversity because we cannot be sure what the past pattern of diversity was in humans. Especially as the mutation distribution density varies between tissue types. A more accurate statement would be that the mutation distribution density in tumours is more likely to resemble humanity's lost diversity than the current patterns in the human germline (though I admit this is an even worse title!). However I think the title would benefit from being changed to something that better reflects the findings of the paper.

Comment on the introduction: I am not sure why the introduction goes into such detail on the role of chromatin state in explaining the distribution of mutations across the genome in tumours when this is just one of many features responsible for the spatial distribution of mutations. The authors also go on to emphasise the tissue-specific variation in mutation distribution across tumours (Lines 47-49). However later on they do not go into detail about how their results vary by tumour type. It

would be interesting if they would comment on how their findings vary by tumour type. Are they uniformly consistent? If not which tissue-types have the strongest and weakest correlations with the human and great ape germline mutational patterns?

Lines 59-59: Could the authors reference their statement that mutation rates seem to have been under selection in the human lineage?

Line 65: Perhaps the authors could provide more information as to their motivation for comparing the mutation distribution in human tumours to germline mutations in the Great Ape lineages? The results are very intriguing but I think it would help the reader if a little more space was given to explaining the logic of deciding to compare these patterns. Especially because the reader may expect tumour mutations may be dominated by non-standard mutational processes (breakdown of normal DNA repair pathways, hypermutation etc). I assume the authors chose tumours because these are a proxy for normal somatic mutation rates but there was a lack of somatic mutation data from these normal tissues, so tumours were chosen as a proxy. However I do not believe this is explicitly stated anywhere in the manuscript.

Line 67: Should it not rather be plural 'the Great Ape Lineages' rather than singular 'the Great Ape lineage'?

Line 70: Why did the authors not also include orang-utans in their study? If such data are available I think it would be good to include them. If the findings are similar to the included NHGA species then the results should strengthen their conclusions and if not it would be interested to address the discrepancies.

Line 113: If mutational density is known to correlate with closed chromatin why is it striking that this was also found in great apes? Is it because this correlation is only found in human tumours and not in the human germline? The authors state in their introduction that there is a correlation in tumours but I do not think they state if the correlation exists in the human germline.

Line 152: Is there any data available from other mammals such as mice that would enable to authors to speculate about just how conserved these patterns might be across mammals? Not essential but it would be of interest to the reader if there is other supporting or conflicting data about this mutational spectrum being present in the germline mutations of other species.

Supplementary Materials:

Lines 345-358: The authors do not mention their motivation for looking at ancestral variation. I assume it is to try to identify the time in the past at which the patterns they observe in extant species appeared. The authors could be clearer in explaining there motivations here. They do discuss this more from line 360 but it leaves the reader a bit confused for the preceding paragraph.

Lines 371-372: Could there also be a role of negative selection, preventing deleterious variants rising to higher frequency?

Lines 399-407: It would be interesting if the authors have any more thoughts on the variation they observe across tumour types, particularly for the tissues with the lowest degrees of correlation. Do they see any patterns of certain tissue types showing higher or lower correlations? It also seems that there is no data on testicular germ cell tumours, which is a shame as here one would presumably expect their observation to be reversed, with human germline and somatic variation patterns correlating more strongly than with NHGA germline mutation rates. I would recommend the authors to include this tumour type if the data is available.

Line 812: What do the authors mean when they say SBS1 has few important components and this

explains the lower correlation with SBS1 compared to SBS5? I don't follow how having fewer components makes a correlation less likely.

Line 852: Typo, 'ad' should be 'and'.

Reviewer #3 (Remarks to the Author): Expertise in great ape genomics

Heredia-Genestar et al. have produced a novel and interesting study that creatively uses data from appropriate non-model organisms in a population genetics and evolutionary context to provide fresh insights into human cancer mutation density. They find that tumor mutation densities more closely resemble the germline mutation densities of other great apes (i.e., gorillas, chimpanzees, and bonobos) than those of humans. They conclude that global losses of nucleotide diversity over human evolutionary history have produced this pattern. Specifically, human genomes have recouped mutations in hypermutable regions (i.e., CpG sites), which have in fact been propagated through rapid, recent population expansion, but remain depleted of other classes of mutations. According to this premise, the mutation densities of other great apes are therefore a better baseline against which to compare cancer mutation densities. Their analyses are thorough, their results overall are well supported, and their interpretation appropriate. Most of the concerns I noted were anticipated by the authors and addressed in the Supplement. I only have minor comments.

I wasn't completely convinced that not mapping to the great ape genomes was a good idea. The authors justify like this "We used the human reference instead of each species own reference to be able to jointly compare Non-Human Great Ape (NHGA) and human samples, and to take advantage of the more complete human gene models and resources available."

To this first point, does this mean that all variants were called jointly? I don't think that would be appropriate for a mixed species/population dataset using the GATK pipeline since genotypes that are more common across samples will be better supported, but I am not sure what else require joint analysis versus using existing annotations, whole genomes alignments like those available for primates on UCSC, and LiftOver? Similarly and to the second point, it seems like reasonable annotations exist for great apes genomes. My concern is that low mappability could then introduce bias, as windows with low mappability are subsequently filtered out. I guess the question would be whether this bias would be greater than the bias that would result from some areas of the great ape genomes being less align-able (some of this would be the same bias, but I think the former would be greater).

The authors mention signatures of selection in passing, but I was curious about how their results might also influence interpretations of signatures of selection, as well the identification of "human accelerated regions."

The identifier "great ape" is not conventionally capitalized.

It might just be the way that the figure was uploaded, but the text on the right-hand side of Figure 3 is cut off.

I thought the high v. low diversity subspecies comparative analysis was really interesting, but might warrant just a little more explication in the main text.

*Line 44: Change "the tumor's" to "tumors"

*Line 90: I think it makes sense to include the test statistic for the chimp-gorilla comparison as the chimp-human and gorilla-human comparisons, in the main text, too.

*Line 104: Could the pattern with the archaics be due to smaller sample size? Maybe not because the authors try to test this for the uneven sample sizes across the great ape subspecies by subsampling the subspecies with larger numbers, but the difference here is that there is a deep temporal separation of these individuals, making the small samples potentially even less representative.

*Line 182: missing period and possible missing last work (germlines)?

*SI Line 130 and 145 - I think missmapping[s] should be mismapping[s] (one s)

*SI Line 175: it seems like SNV should be plural (SNVs)

Reviewers' comments:

Reviewer #1 (Remarks to the Author): Expertise in comparative genomics

In the manuscript, “Mutation distribution density in tumors reconstructs human’s lost diversity”, the
authors Heredia-Genestar et al. map the distribution of mutational density in: 1) ~4,000 humans (1000
Genomes project + Simons Diversity Panel); 2) Non-human great apes (69 chimpanzees/bonobos +
43 gorillas); 3) three archaic humans and 4) ~2,500 human tumors (Pan-Cancer Analysis). They then
correlate these mutational densities against each other, and find that non-human great apes have a
significantly higher correlations with human tumors than the human samples tested – including the
archaic samples. The authors then test the distribution of non CpG vs CpG>T and find the density of
non CpG in NHGA and human tumors to be more correlated than humans. Additionally, they confirm
that germline mutations rates in humans have undergone a recent slowdown. Authors suggest that
humans do not have the expected mutation densities may be due to losses in population diversity from
bottlenecks and expansions.

Overall, I find this to be a very careful and overall thorough study, with substantial additional testing
in the supplement, including various sizes of sliding widow for analyses, additional datasets, subsets
of humans, subsets of tumors, and variant frequency - suggesting the signal the authors are finding is
indeed true. I also find the methods and results to be solid and an important contribution to the field of
human evolution and cancer biology. I have a few concerns and questions about the interpretation and
potential additional analyses outlined below.

We appreciate the positive comments of the reviewer and her/his recognition of the solidity of our
methods and results.

Major/Minor Concerns:

1. The underlying theory, motivation and hypothesis for testing these distributions is a bit unclear in
the introduction.

We thank the reviewer for pointing this out. We have clarified this point in the Introduction (lines 76-
88).

2. I have a hard time with the interpretation of the data.

We agree that this is a complex subject, but we have improved the manuscript to provide more
context on each analysis and to better connect the multiple analyses.

While I believe that the results are true, it's unclear the mechanism of how these tumor mutation
distributions are correlated more closely to NHGA than humans.

As now it is better explained in the revised manuscript, CpG>T and non-CpG mutation density
landscapes are decoupled in the human germline (they are poorly correlated). This decoupling is
human-specific and seems to be associated with the lower correlation of human non-CpG mutation
densities with the other datasets. We observed that the lower correlation of non-CpG mutation
densities is common in low diversity populations, but decoupling seems to be a human-specific
phenomenon.

We propose that this decoupling results by a combination of the complex demographic history of
humans and the previously described slowdown in human mutation rates. Multiple population
bottlenecks in human populations would have reduced diversity in population-specific ways. Recent
amplification of some of these populations would have skewed the genome-wide distribution of
mutation densities. Mathieson, Plos Genetics, 2017 reported a similar finding in human Native
American populations, although they did not analyze mutation distribution. We propose that the
different behavior of CpG>T and non-CpG mutations is explained by a combination of a more
regionally constrained background for CpG>T mutation driven by CpG-content (observed in the small
set of *de novo* mutations) and the previously described slowdown of the mutation rates in humans
impeding the recovery of the expected distribution of mutation densities.

I wonder if the results are highlight different levels of DNA repair more than the actual mutation
burden. It may be that human tumors and NHGA have similar mechanisms of repair.

The reviewer raises a very interesting point. Our initial hypothesis was that changes in the genome-
wide distribution of the contribution of the different sources of DNA damage (including repair
mechanisms) might explain the human-specific behavior. To discern this we introduced the

trinucleotides analysis. However, we couldn't find species-specific differences in trinucleotides
known to be associated with any mutagen or repair machinery.

What we could detect, however, was an association between the individual tumor type burden of
mutations associated with signatures SBS5 and SBS40, with a higher tumor-NHGA similitude in
multiple trinucleotides. These two signatures, however, are not associated with any known mutational
process. Quite the opposite, these signatures are associated with normal cell processes and mutation
accumulation in healthy tissues.

As explained above, we propose that the human-specific slowdown of the mutation rates contributes
to the inability of non-CpG mutations to recover their expected landscape. This slowdown could be
generated by a general higher efficiency of DNA repair machinery, although specific differences in
spermatogenesis/oogenesis could lead to similar results. These effects, however, would be very
difficult to detect just by the analysis of mutation spectrum, as they would only cause a reduction of
the number of mutations per year, without causing shifts in the mutation spectra. We have
incorporated a brief discussion of this point in the Discussion (although for the sake of clarity we
haven't commented on the specific alternative scenarios).

Additionally, tumors have accelerated mutations and distributions, so I wonder if this relationship
would hold with all species with faster germline mutation rates.

We agree with the reviewer. Mutation rates must have an influence on the ability of any evolving
population to show the level of diversity needed to reflect the expected distribution of mutation
densities. Actually, we think that the high slowdown of the mutation rates in human contributes to the
difficulty of human populations to recover their expected genome-wide landscape. In brief, the
population diversities of chimpanzees and gorillas (which are not fast-evolving species) would be
mimicked by the accumulation of mutations in tumors with high mutations rates but not in the human
population with low mutation rates. It would be interesting to explore if other closely related species
with different mutation rates and population histories show similar phenomena (reflecting their own
landscapes). Another interesting future development would be the analysis of NHGA tumors, as other
systems with high mutation rates, although no such data is currently available. We have rewritten the
manuscript to better reflect the combined contribution of human-specific changes in mutation rates
and population evolution (main text lines 191-214; 333-349).

3. I have a few questions about the samples/data.

a. What were the male/female distribution in the samples – both human, NHGA and tumors?

We thank the reviewer for noticing that this information was missing, the Great Apes datasets are
female-biased (Male/Female samples, 1kGP: 1,233/1,271; chimpanzee: 23/45; gorilla: 10/33; tumor:
1,420/1,160). However, our analyses use only autosomes. The samples in the germline datasets show
no difference in heterozygous mutational load per sample between sexes in their autosomes (as
expected, as the immense majority of mutations present in all cells of the individual are inherited from
both the father and the mother through their gametes, before fecundation, regardless of the sample's
sex).

We divided the tumor datasets by sex and observed the same patterns described when using all
samples. As some tumor types are sex-specific, we repeated this analysis removing breast, uterine,
cervix, ovarian, and prostate cancers, and obtained identical results (we haven't included this specific
result in the manuscript for the sake of simplicity).

We have included a new section in the supplementary text (lines 449 – 472) and Supplementary Table
4c detailing these analyses about the influence of sex and age (next question) in our analysis.

b. Were any of the subsets age-matched to see if the pattern holds? Mutation signatures likely
correlate with age, and likely many of the tumors represent an older population of individuals. How
do ages reflect in the human and NHGA data?

This is an excellent question. There is no phenotypic information about age from the 1kGP (due to
sample privacy) and NHGA samples (majority of wild-born samples with unknown age). Moreover,
the “germline” genomes are actually from “somatic blood samples”, and mutation load in normal
whole-blood is highly correlated with sample age (García-Nieto, *Genome Biology*, 2019). However,
as the total number of somatic mutations in blood progenitors is relatively small (~1,000 at age 60; 14
115 per year) (Osorio, *Cell Reports*, 2018) compared with the 1~2M heterozygous sites per human or
116 NHGA sample, sample age in germline datasets should have a negligible effect in our study.

Cancer incidence and tumor mutation load do in fact correlate with the sample's age. This correlation
is mostly driven by the increased number of cell replication cycles with increased age, even in healthy
tissues (Martincorena, *Science*, 2018). In addition, different tissues replicate at different speeds. The
combination of these two effects can be observed in Supplementary Table 4a, with slow-replicating
tissues like bone and brain having a low number of mutations even when having a similar or higher
number of samples than other fast-replicating cancer types. However, as suggested by the reviewer
this effect can be influenced by the sample's age, as (eg. the brain cancer includes pediatric brain
cancers with low mutation load).

To address the possible effect of patient age in tumors, we have divided the tumor dataset into 3
groups: 0-29 years, 30-59 years, and 60-99 years. The old age group included most samples (1,328
samples) and the highest number of mutations (19.7M). For the three tumor age groups, NHGA-tumor
correlations are higher than human-tumor ones. Consistently with the smaller number of samples and
mutations, the youngest group shows slightly weaker correlations, similar to the correlations observed
for tumor types with a comparable number of mutations.

We have included a new section in the supplementary text (lines 449 – 472) and Supplementary Table
4c about the influence of sex (previous question) and age in our analysis.

4. For the tumor subset analyses – would the authors expect cancer from germline mutations to have a
better correlation between human mutation distribution than NHGA and somatic mutation tumors?

In principle, we would agree with the reviewer. We would expect cancers arising from the human
germinal line (Testicular Germ Cell Tumors (TGCT) and Ovarian Germ Cell Tumors (OGCT)) to
have stronger correlations with the mutation distribution present in healthy human samples from a
population (what we called “germline”), and even with NHGA samples. Although the extent of these
differences with somatic mutation tumors is difficult to foresee.

Unfortunately, the original PCAWG tumor dataset included no Germ Cell Tumors (the Ovarian-
Adenocarcinoma tumor type arises from epithelial cells in the ovary). Instead, we obtained two
additional datasets of Whole-Genome Sequenced TGCT: Spermatocytic tumors (Giannoulatou, Plos
One, 2017), and Nonseminomatous Testicular Germ Cell Cancer (Dorssers, British Journal of Cancer,
2019). Spermatocytic tumors present mutation patterns similar to those of human populations, as they
arise from the same germinal cells precursors of spermatocytes. However, these tumor types have
very low mutation rates, and the number of mutations provided by these two studies is not enough to
properly assess the mutation distribution across the genome.

We now present these additional analyses in the supplementary text (lines 422-447) and in
Supplementary Table 4a.

5. NHGA have higher SNVs due to between subspecies differences – could this be one possible
explanation for the correlation between cancer & NHGA?

As the reviewer points out, in principle, inter-species differences when pooled could result in
similar distributions to inter-tumor differences when considered together. To address this, we
analyzed both tumor-types and NHGA subspecies separately.

For tumors, we found that virtually all tumor-types) show higher correlations with NHGA than with
human (see lines 412-420 in Supplementary Text and Supplementary Table 4a).

In the analysis of NHGA subspecies we observed that high-diversity subspecies also present high
correlations with tumors (Supplementary Table 3a). In these subspecies analysis, we used only
segregating sites within the subspecies, removing all fixed variants within that subspecies (we now
have clarified this point in Supplementary Text lines 509-511). We have rewritten the Subspecies'
diversity section in Results (lines 125-148 of the main text) to further clarify these results.

6. In the supplement, authors mention regions with low homology between humans and NHGA would
show extremely low mutation density in chimpanzee and gorilla because we would not be able to call
variants there. I am wondering if this could bias the results and influence how and where cancer
mutations land in the genome.

Here the reviewer raises a potentially important point. Actually, during these analyses we were also
concerned by the possible influence of focusing on those regions where reads from NHGA could map
reliably in the human genome.

First of all, all samples from all datasets are mapped and called against the whole human genome
(hg19), and regions are filtered out only after the variant calling is finished. Consequently, tumor (and
human) mutations mapping to these low homology regions are simply removed and are not mapped to
other regions of the genome. The main filter affecting these regions, however, is the mappability
mask, where we cut the human reference into 35-bp k-mers and map them against the same reference.
This detects regions where reads do not map uniquely and can cause mismappings in mapping reads
either from NHGA and tumors. This mask is independent of NHGA and removes ~19.78% of the
non-N genome, and a similar mask has been applied to the release of the 1kGP dataset. Additionally,
the CNV and Callability in NHGA filters remove an additional 3.38% of the non-N genome.

To address this point, we repeated the analyses using different filtering thresholds (Supplementary
Table 2a and Supplementary Figure 1b). We observed the same distribution patterns and behavior of
correlations in all the cases, regardless of the strictness of these filters. Additionally, we analyzed the
correlations using an unfiltered version of the datasets (removing only windows with $\geq 75\%$ of the
window missing due to centromeres, telomeres, and such) obtaining very similar correlations to the

filtered data: human-tumor=0.20; chimpanzee-tumor=0.55; gorilla-tumor=0.60. Consequently, we
showed that the higher NHGA-tumor than human-tumor correlation is robust to the filtering of
problematic regions.

7. Lastly, the conservation of mutation spectra in additional species, including vervet monkey – I was
confused on whether the tumor samples showed stronger correlation with human tumors than
humans?

We thank the reviewer for pointing to this unclear point. All the mutation spectra used in the
manuscript are calculated on germline population data, but not in tumors. We clarified this in the
Main Text (line 220-221).

The mutation spectra from tumors varies wildly from one tumor type to the other due to the
mutational processes acting in the tumor. When combining all tumor types, the aggregate mutation
spectrum of all tumors is very different than any germline datasets (Pearson's $R = 0.36-0.39$ across
Great Apes).

We included the correlations with tumor mutation spectra in Supplementary Table 10c.

Reviewer #2 (Remarks to the Author): Expertise in comparative genomics (cancer)

I found this to be a creative manuscript with novel analyses and enjoyed reading it. The main
observation is quite striking, that there is a stronger correlation in the mutational rate across the
genome between human tumours and NHGA germline than there is between human tumours and the
human germline. This provides new insights into differences between somatic and germline mutations
and about changes in mutational rates and patterns across the great ape lineages. The authors speculate
about the causes for these observations, with the most likely explanation being past changed in human
population size resulting in a decrease in diversity followed by a relative increase in the accumulation
of more clock-like CpG > T transitions. I believe this observation is a useful contribution to the field
of genomics and will be of interest to those working in the fields of both somatic and germline
evolution.

We thank the reviewer for their positive feedback of our work. Especially, we appreciate the extensive
review and the interesting suggestions that we believe have improved considerably the manuscript.

I have two main comments. The first is that the flow of the results section of the manuscript is
sometimes difficult to follow and it is not always clear why certain analyses were done. I think adding
a few additional sentences before some analyses explaining the motivations could help guide the
reader. Having said that the discussion section does a good job of summarising the key findings.

We thank the reviewer for this suggestion. We have rewritten extensively the manuscript to better
clarify the motivation of the analyses as well as the flow of the text. Some specific changes include:

- ● Main text lines 41-54: Improved the introduction to better summarize some previous studies
pointing to the “normal” mutational behavior of tumors.
- ● Main text lines 76-88: Improved the final part of the introduction motivating our study based
on the need to reconcile previous observations on the mutational “peculiarities” of the human
germline, the “not so peculiar” behavior of tumors and the strong differences between both
systems.
- ● Added sub-headings in the results section to better space the analyses.
- ● Main text lines 126-128: Expanded the rationale behind the subspecies analysis
- ● Main text lines 151-158: Expanded the rationale behind the chromatin status and genomic
features analyses.
- ● Main text lines 174-214: Rewriting of the analyses of CpG>T and non-CpG mutations to
better explain the observed behavior in all the datasets.

- ● Main text lines 217-219: Better formatted the rationale behind the whole-genome mutation
spectra analysis
- ● Main text lines 241-245: Better differentiated the whole-genome mutation spectra analysis
from the trinucleotide window distribution across the genome analysis to identify active
mutational signatures.

We really appreciate the reviewer's comment on the discussion and driven by the clarifications
introduced in results we have updated the paragraphs discussing the possible phenomena causing the
detected behavior (Main text line 305-349) and have rewritten the last paragraph and added two final
paragraphs to better discuss some implications of our results (Main text line 351-372).

The second main comment is that currently the manuscript presents a striking observation and
speculates on the causes without doing any further analyses or experiments to explore them. I don't
think this is the fault of the authors as it would probably be very difficult to think of ways to test
which explanation for the observations is correct. However if it were possible to do any experiments
such as simulation studies of human population size changes and explore whether these do indeed
recapitulate the patterns they observe relative to great apes, that would certainly strengthen the paper.
However I recognise that this would add substantial work and could probably in itself be a separate
manuscript.

We thank the reviewer for the suggestion. We fully agree that we present several key observations
which causes are a matter of discussion. We have been able to discard the importance of some
obvious possible explanations (such as drastic changes in mutation-generating phenomena) and have
replicated some important observed behaviors (like the effect of the loss of diversity). However, we
agree that our current manuscript leaves open the important question of which is the cause of the
human-specific decoupling of CpG>T and non-CpG mutation densities. We discuss the possible
impact of the combination of population expansions and mutation slowdown, but testing of this
hypothesis is left for future investigations. Interestingly, there is a recent precedent of a related
observation briefly mentioned in our manuscript (main text line 320) by Mathieson et al, as they
detected signals of positive selection or mutation acceleration in CpG>T mutations in Native
Americans deemed to be caused by population expansion.

We agree that a simulation analysis would be the best way to undertake this kind of analysis.
However, as the reviewer comments this is a really complex study that would be better fitted in a
separate manuscript and is out of the scope of this work. Some of the considerations that such a
simulation should/might fulfill would be:

- 1. Simulate a population of whole-chromosome sequences or even whole genomes (to be able to
analyze the mutation distribution in 1Mbp windows) with different context-dependent
mutation rates for each of the 96 trinucleotide contexts (or at least CpG vs non-CpG sites)
properly managing their location in the genome.
- 2. Properly simulate both increases (human population expansions) and decreases in population
size (transition from the high-diversity of human-chimp common ancestor to low-diversity
human “bottlenecked” populations). These expanding/decreasing population simulations
should be applied also to NHGA subspecies to simulate their history of bottlenecks with no
following expansion.
- 3. It should be able to combine sequences of bottlenecks and expansions in different
subpopulations within each species to generate complex population structures (population
admixture could also be considered at some point).
- 4. Properly simulate recombination, as it is both mutagenic and higher in regions of high CpG
sites.
- 5. It should be able to tune the contribution of different mutational signatures to simulate
changes in mutation rates either globally or in a global way.
- 6. This model should be able to simulate the effect of clock-like Vs replication-dependent Vs
mutagen-dependent mutations.
- 7. Changes in longevity and/or generation time should be simulated as they are a possible source
of differences in mutation rates between clock-like and replication-dependent mutations.
- 8. Influence of (epi)genomic features could be also included to determine their effect on the
differences found between germline and somatic mutations (provided that we were able to
accurately define the germline epigenetic landscapes in every species).

We think that developing a simulation system like this would require an important amount of work
and is out of the scope of this work. Moreover, performing the simulations with the combinations of
parameters compatible with the current state of knowledge (sometimes contradictory and often
uncomplete) is clearly another (quite demanding) project deserving its own publication.
Notwithstanding, we fully agree that this is a very interesting project that has the potential for helping

to understand the phenomena driving the evolution of different systems while reconciling some
paradoxical observations like the one reported in this manuscript.

Below are some minor comments.

Main Text:

Line 2: Technically I think the word ‘human’s’ would need to be changed to ‘humanity’s’ or ‘human-
kinds’ to be grammatically correct. Also the mutation distribution density in tumors may not
reconstruct humanity’s lost diversity because we cannot be sure what the past pattern of diversity was
in humans. Especially as the mutation distribution density varies between tissue types. A more
accurate statement would be that the mutation distribution density in tumours is more likely to
resemble humanity’s lost diversity than the current patterns in the human gremlin (though I admit this
is an even worse title!). However I think the title would benefit from being changed to something that
better reflects the findings of the paper.

We thank the reviewer for pointing out the inconsistencies of the original title. We suggest a new title
with:

Extreme differences between human germline and tumor mutation densities are driven by ancestral
human-specific deviations.

Comment on the introduction: I am not sure why the introduction goes into such detail on the role of
chromatin state in explaining the distribution of mutations across the genome in tumours when this is
just one of many features responsible for the spatial distribution of mutations. The authors also go on
to emphasise the tissue-specific variation in mutation distribution across tumours (Lines 47-49).
However later on they do not go into detail about how their results vary by tumour type. It would be
interesting if they would comment on how their findings vary by tumour type. Are they uniformly
consistent? If not which tissue-types have the strongest and weakest correlations with the human and
great ape germline mutational patterns?

We appreciate the reviewer’s suggestion. We have expanded the introduction to mention the effects of
replication time and the recruitment of MMR on mutation distribution, as well as expanding the

concept of using tumors as a proxy of somatic mutation that you suggested in another comment (see
below).

We also addressed (Main text lines 151-158) the results of the correlations per individual tumor type
(the human-NHGA-tumor pattern holds for all tumor types) using it as a connection with the
chromatin status analysis. We have also expanded the analysis of individual tumor types in the
supplementary (Supplementary Text lines 422-447), analyzing more closely the correlations between
the most mutated tumor types.

Among the tumor types with more than 1M SNV in the dataset, we detect very high correlations in all
but two types: Melanoma and Breast Adenocarcinoma. Melanoma mutation distribution is strongly
affected by mutational signature SBS7a-d (affecting TCx>T sites) and has been associated with the
nuclear periphery (García-Nieto, EMBO J., 2017), suggesting potential causes for the differentiation.
The differences in the mutation distribution of Breast Adenocarcinoma have, to our knowledge, not
been previously described. We hypothesized that this differential pattern might be caused by samples
with protein-truncating variants in the *BRCA1* and *BRCA2* genes (Waszak, biorXiv, 2017; The
ICGC/TCGA Pan-Cancer Analysis of Whole Genomes Consortium, Nature, 2020). Thus, we split the
dataset into two groups, with samples presenting or not mutational signature SBS3. However, both
groups showed identical correlations with the other tumor types, suggesting that *BRCA1/2* deficiency
is not the cause behind the Breast Adenocarcinoma different mutation distribution.

Lines 59-59: Could the authors reference their statement that mutation rates seem to have been under
selection in the human lineage?

Corrected. Now in Main text line 70.

Line 65: Perhaps the authors could provide more information as to their motivation for comparing the
mutation distribution in human tumours to germline mutations in the Great Ape lineages? The results
are very intriguing but I think it would help the reader if a little more space was given to explaining
the logic of deciding to compare these patterns. Especially because the reader may expect tumour
mutations may be dominated by non-standard mutational processes (breakdown of normal DNA
repair pathways, hypermutation etc). I assume the authors chose tumours because these are a proxy
for normal somatic mutation rates but there was a lack of somatic mutation data from these normal

tissues, so tumours were chosen as a proxy. However I do not believe this is explicitly stated
anywhere in the manuscript.

Actually, we think this is an important point for improving the clarity of the manuscript. The reviewer
is right, we used tumors as a proxy of somatic mutations. Now, we have clarified our motivation to
understand the contribution of human-specific mutational patterns in human-tumor differences. This
motivation comes from the convergence of previous studies suggesting that most mutations in tumors
reflect “normal” somatic patterns and the description of human-specific mutational behaviors (as the
mutation slowdown), suggesting that germline mutations in human are different from other great apes.

Line 67: Should it not rather be plural ‘the Great Ape Lineages’ rather than singular ‘the Great Ape
lineage’?

Thank you for the correction. The final part of the discussion has been completely revamped and this
sentence no longer exists.

Line 70: Why did the authors not also include orang-utans in their study? If such data are available I
think it would be good to include them. If the findings are similar to the included NHGA species then
the results should strengthen their conclusions and if not it would be interested to address the
discrepancies.

It is interesting that the reviewer raises this point. At the inception of this project, the number of
orangutan samples was very small and with low coverage, and we decided not to include orangutan in
the analysis. However, new orangutan population data is available and we have now included their
analysis in a separate section (Main text 121; Supplementary Text 549-570). Including them in the
main human-chimpanzee-gorilla-tumor analysis would imply rebuilding all filters and analysis.

We analyzed 24 orangutan samples and observed the same correlation dynamics in orangutan as in
other NHGAs. However, the diagonal pattern is weaker in orangutan than in other NHGAs, mainly
caused by a weaker human-orangutan correlation. This suggests that although the strong NHGA-
tumor correlation is present also in orangutan, the specific diagonal pattern depends on the distance
between species. This effect of the evolutionary divergence is also observed in a weaker Mann-
Whitney U test p-value for gorillas than for chimpanzee, and in the lower human-gorilla correlation
than the human-chimpanzee one.

Line 113: If mutational density is known to correlate with closed chromatin why is it striking that this
was also found in great apes? Is it because this correlation is only found in human tumours and not in
the human germline? The authors state in their introduction that there is a correlation in tumours but I
do not think they state if the correlation exists in the human germline.

The reviewer is right. There is a clear association between chromatin state and tumor mutation density
that is missing (or much weaker) in the human germline (Figure 2 in this manuscript; Figure 1 in
Schuster-Böckler, Nature, 2012). In fact, in the context of our results this lack of correlation is
consistent with its inability to reflect the expected distribution of mutation densities. This is now
stated in Results (line 166) and in Discussion (lines 270-272).

Line 152: Is there any data available from other mammals such as mice that would enable to authors
to speculate about just how conserved these patterns might be across mammals? Not essential but it
would be of interest to the reader if there is other supporting or conflicting data about this mutational
spectrum being present in the germline mutations of other species.

Milholland, Nature Communications, 2017, analyzed germline and somatic mutation rates in humans
and mice. They detected faster mutation rates in the soma compared with the germline in both species.
The mice mutation rate (both germline and somatic) was higher than the human mutation rate. Both
species presented unique germline and somatic mutation spectra, with CpG>T mutations being
predominant specifically in the human germline. This result matches our observation of a lower
correlation between the mutation spectra of primates and mice (Supplementary Table 10c), and a low
correlation between the mutation spectra of mice and the mutational signature SBS1 (CpG>T). We
have expanded the description of the conservation of these patterns the main text (lines 228-231).

Supplementary Materials:

Lines 345-358: The authors do not mention their motivation for looking at ancestral variation. I
assume it is to try to identify the time in the past at which the patterns they observe in extant species
appeared. The authors could be clearer in explaining there motivations here. They do discuss this
more rom line 360 but it leaves the reader a bit confused for the preceding paragraph.

We have added a brief motivation for the analysis at the beginning of the section (Supplementary Text
lines 475-477).

Lines 371-372: Could there also be a role of negative selection, preventing deleterious variants rising
to higher frequency?

That could be an interesting possibility. In the referenced article (Carlson, Nature Communications,
2018), the authors compare the BRIDGES dataset (extremely rare variants, very recent in time) with
intergenic variants from 1kGP of multiple allele frequencies. The authors detect slight differences in
the mutation frequency of certain sequences. However, the nature of the compared datasets
(autosomic vs intergenic) does not allow inferring the effect of selection. However, the changes in
frequencies suggest a more global trend compatible with the effect of mechanistic differences (maybe
driven by negative selection). In principle, although it could also result from the aggregated effect of
an increase of negative selection in many places, it seems to be a less parsimonious explanation.

Lines 399-407: It would be interesting if the authors have any more thoughts on the variation they
observe across tumour types, particularly for the tissues with the lowest degrees of correlation. Do
they see any patterns of certain tissue types showing higher or lower correlations? It also seems that
there is no data on testicular germ cell tumours, which is a shame as here one would presumably
expect their observation to be reversed, with human germline and somatic variation patterns
correlating more strongly than with NHGA germline mutation rates. I would recommend the authors
to include this tumour type if the data is available.

We thank the reviewer for these interesting suggestions. We have further developed this analysis
(Supplementary Text lines 422-435). For a discussion about differences between tumors, see our
answer to the second minor comment of the reviewer.

We really appreciate the reviewers' suggestion of including the germ cell cancers. In fact, we would
expect human-tumor correlations improve in these cases, but the extent of this improvement (and
whether they would revert our main observation) is difficult to foresee.

Although the original PCAWG dataset does not include any germ cell tumors, we could retrieve data
from two studies on testicular germ cell tumors: spermatocytic tumors (Giannoulatou, Plos One,
2017), and nonseminomatous Testicular Germ Cell Cancer (Dorssers, British Journal of Cancer,
2019). Unfortunately, we could not obtain whole-genome data from ovarian germ cell tumors.

Spermatocytic tumors have been previously associated with mutation signatures SBS1 and SBS5,
similar to the human germline (higher in SBS1 due to the increased CpG methylation of testes)
suggesting that they could be more similar to human germline than other tumors. However, these

tumor types have a low number of mutations even after combining them (2,710 SNV across 8
samples) and did not afford enough power for our mutation distribution analyses (Supplementary
Table 4a). Future work including larger datasets will be necessary to answer this interesting question.
This additional analysis is now explained in Supplementary Text (lines 437-447).

Line 812: What do the authors mean when they say SBS1 has few important components and this
explains the lower correlation with SBS1 compared to SBS5? I don't follow how having fewer
components makes a correlation less likely.

The point we try to explain is that SBS1 is driven by a small number of mutation types (almost
exclusively xCG>T; <https://cancer.sanger.ac.uk/cosmic/signatures/SBS/SBS1.tt>). This situation
contrasts with the case of SBS5 which has a contribution of all the components
(<https://cancer.sanger.ac.uk/cosmic/signatures/SBS/SBS5.tt>). These differences imply that even
although CpG>T mutations have a disproportionately high contribution to human mutations, the
correlations of SBS1 with all the mutations spectra is lower than those of the most comprehensive
SBS5 (even although each of the mutation types have a lower contribution to the global landscape).
This highlights the difference between mutation-focused and global spectra comparisons

Line 852: Typo, 'ad' should be 'and'.

Thank you for the correction. Corrected in Supplementary Text (line 949).

Reviewer #3 (Remarks to the Author): Expertise in great ape genomics

Heredia-Genestar et al. have produced a novel and interesting study that creatively uses data from
appropriate non-model organisms in a population genetics and evolutionary context to provide fresh
insights into human cancer mutation density. They find that tumor mutation densities more closely
resemble the germline mutation densities of other great apes (i.e., gorillas, chimpanzees, and bonobos)
than those of humans. They conclude that global losses of nucleotide diversity over human
evolutionary history have produced this pattern. Specifically, human genomes have recouped
mutations in hypermutable regions (i.e., CpG sites), which have in fact been propagated through
rapid, recent population expansion, but remain depleted of other classes of mutations. According to
this premise, the mutation densities of other great apes are therefore a better baseline against which to
compare cancer mutation densities. Their analyses are thorough, their results overall are well
supported, and their interpretation appropriate. Most of the concerns I noted were anticipated by the
authors and addressed in the Supplement. I only have minor comments.

We appreciate the positive comments of the reviewer about our work.

I wasn't completely convinced that not mapping to the great ape genomes was a good idea. The
authors justify like this "We used the human reference instead of each species own reference to be
able to jointly compare Non-Human Great Ape (NHGA) and human samples, and to take advantage
of the more complete human gene models and resources available."

To this first point, does this mean that all variants were called jointly? I don't think that would be
appropriate for a mixed species/population dataset using the GATK pipeline since genotypes that are
more common across samples will be better supported, but I am not sure what else require joint
analysis versus using existing annotations, whole genomes alignments like those available for
primates on UCSC, and LiftOver? Similarly and to the second point, it seems like reasonable
annotations exist for great apes genomes. My concern is that low mappability could then introduce
bias, as windows with low mappability are subsequently filtered out. I guess the question would be
whether this bias would be greater than the bias that would result from some areas of the great ape
genomes being less align-able (some of this would be the same bias, but I think the former would be
greater).

We appreciate the reviewer's concern on a possible source of bias in the data. This is a complex issue
that we will address point by point.

1- The samples were not called jointly. They were called individually using GATK HaplotypeCaller
generating an intermediate gvcf file. All individual samples within a species (chimpanzee, gorilla, and
sgdp_50 separately) were then jointly called following GATK best practices. We explained this more
clearly in Supplementary Text (lines 112-114).

2- We agree that mapping samples to their species reference genome should provide a better quality
callset with lower FDR (provided the assembly has enough quality). However, this will introduce the
additional bias of dealing with different quality assemblies and requires the additional step of the
mapping between references. Importantly, mapping Great Ape samples and populations to the human
reference results also in high-quality datasets and has been used in several Great Ape flagship projects
(Prado-Martinez, Nature, 2013; De Manuel, Science, 2016).

3- Our analyses require all datasets to share the same set of coordinates. Here, we are not only
interested in having high-quality SNV calls, but also in properly assessing the mutation distribution in
windows across the genome. To do this we need an unbiased distribution of mutations across
windows in the chosen reference. We made this clearer in Supplementary Text (lines 77-84).

4- The callability filter is unavoidable, regardless of the reference genome used. This filter is based on
the quality of the samples. As NHGA samples were generated in different sequencing centers, they
present some degrees of batch effects in their quality. The usage of this filter allows us to identify
regions of the genome (in whatever reference used) where $\geq 25\%$ of the species' samples had poorer
ability to call variants. The callability filter of 50 chimpanzee samples in hg19 and panTro5 is as
follows (#bp passing the filter):

Callability 50 chimp hg19 (autosomes): 2,375,256,148

Callability 50 chimp panTro5 (autosomes): 2,438,687,055

difference: ~60 Mbp (panTro5 > hg19)

5- The mappability filter is also unavoidable, as it masks the reference genome removing regions that
do not map uniquely to the same reference, resulting in potential mismappings of reads leading to
increased or decreased mutation densities. The hg19 reference has a higher number of bp mapping
univocally and passing this filter (hg19: 2,170,617,569 bp ; panTro5: 2,140,243,942 bp in autosomes;
difference ~30 Mbp hg19>panTro5).

6- When combining (intersecting) both the mappability and callability (measured in 50 chimp
samples) filters, there is slightly more sequence passing the filters in hg19 (hg19: 2,098,608,751 bp ;
panTro5: 2,077,115,584 bp in autosomes ; difference ~20 Mbp hg19>panTro5).

7- The use of liftOver to re-coordinate chimpanzee samples from panTro5 to hg19 is an interesting
 suggestion. We compared the results of mapping and calling 50 chimpanzee and 50 human samples in
 both the hg19 and panTro5 references. We also used liftOver to transform these callsets to the other
 reference (hg19 to panTro5 and vice-versa). We then compared the two versions of the dataset (50
 chimpanzee mapped to panTro5 vs 50 chimpanzee mapped to hg19 and liftOver-ed to panTro5, etc.)
 obtaining the variant results in the same genomic location in both datasets. We compared the total
 number of non-fixed variants called in the pre-liftOver dataset in autosomes, the number of non-fixed
 variants able to liftOver to the destination reference and the number of non-fixed variants able to
 liftOver but wrongly called (mismatching REF/ALT alleles in both datasets, with allele frequency >
 singletons). The results are shown in the following table:

	chimp hg19->panTro5	chimp panTro5->hg19	sgdp_50 hg19->panTro5	sgdp_50 panTro5->hg19
#nonfixed SNV called	30,749,336	33,220,639	14,460,298	13,269,827
#nonfixed SNV able to liftOver (% over total called)	30,435,834 98.98%	32,976,410 99.26%	14,349,954 99.24%	13,194,569 99.43%
#nonfixed SNV able to liftOver but not properly matching (% properly called over total called)	624,255 96.95%	583,295 97.51%	388,826 96.55%	73,178 98.88%

We obtain very similar numbers in both human and chimpanzee regardless of the direction of the
 comparison. Although the number of variants called is slightly higher in the datasets using their own
 species' reference, both mapping directly or mapping+liftOver result in very high levels of
 coincidence in the results of their calling. This suggests that both methods give similar results in the
 variant calling.

8- However, although both direct mapping and mapping+liftOver give similar results in the variants
 called, there are important differences in their distribution. Using liftOver from panTro5 to hg19
 generates some “dark spots” in the genome where no SNVs are mapped to. These “dark spots” are
 caused by a combination of regions where liftOver cannot map to (none of the liftOver
 panTro5Tohg19 chains directs to that position), and regions filtered out by the mappability+callability
 filter in the panTro5 calling. These effects persist even after filtering the resulting data using the hg19
 version of the mappability+callability filter. This can be observed in the following figure when
 comparing the ranked distribution of windows in the datasets mapped directly to hg19 (x-axis) or
 mapped to panTro5 and liftOver-ed to hg19 (y-axis):

The points below the diagonal represent windows, in the same dataset using the exact same samples,
 that have lower mutation density in the liftOver panTro5->hg19 dataset than in the directly mapped
 dataset. This effect is caused solely by liftOver re-mapping, as the affected windows are shared by
 both chimp_50 and sgdp_50 datasets (data not shown). These windows would have an effect on our
 inter-species estimates of mutation density.

9- Importantly, although we use a single reference genome to run these analyses, we can detect
 identical patterns when using either hg19 or panTro5 genomes (Supplementary Figure 1h).

The authors mention signatures of selection in passing, but I was curious about how their results
 might also influence interpretations of signatures of selection, as well the identification of "human
 accelerated regions."

Certainly, this is a very important point that deserves further future work. Interestingly, Mathieson et
 al (Mathieson, Plos Genetics, 2017) detected an apparent signal of selection associated with increased
 mutation rates at CpG>T sites caused by recent population expansion leading to an increased allele
 frequency of CpG>T sites. As CpG>T sites are the most common mutation in humans, in a population
 expansion scenario these mutations arise faster than other types of mutations and can reach higher
 non-singleton allelic frequencies. This drift-base effect will generate false signatures of positive
 selection.

Our analysis suggests that these effects are not limited to specific populations and have been active
 through human's recent evolution. In addition, the higher CpG>T mutation rate affects more CpG
 sites. These false signatures of positive selection can show a higher presence in CpG-rich exons.
 Another possibility is that CpG>T deleterious variants can arise more frequently due to this same
 effect.

As for the identification of human accelerated regions, we would suggest the removal of CpG sites
from the analysis. However, CpG>T mutations co-localize with PRDM9-associated recombination
hotspots in human males. Recombination is mutagenic and its hotspots are associated with GC-rich
regions. Moreover, recombination-induced double-strand breaks are associated with CpG>T
mutations in the male germline, and non-CpG, C>T mutations in the female germline (Halldorsson,
Science, 2019). This suggests that, although CpG-rich regions are inherently accelerated because of
the higher mutation rate, not all the mutation acceleration is caused by CpG sites and these regions
should be carefully considered.

We have expanded the discussion of this point in Main Text (lines 320-325).

The identifier “great ape” is not conventionally capitalized.

We thank the reviewer for calling our attention to this mistake. We have corrected this across both the
Main Text and Supplementary Text.

It might just be the way that the figure was uploaded, but the text on the right-hand side of Figure 3 is
cut off.

We thank the reviewer for pointing out this problem. We reformatted the figure and uploaded an
updated version.

I thought the high v. low diversity subspecies comparative analysis was really interesting, but might
warrant just a little more explication in the main text.

We appreciate the reviewer’s interest and we have expanded the rationale of the analysis and added
more information in the main text (lines 126-132). We think that this also improves the clarity of the
manuscript.

*Line 44: Change “the tumor’s” to “tumors”

Thank you for finding this typo. Corrected in the Main Text (line 45).

*Line 90: I think it makes sense to include the test statistic for the chimp-gorilla comparison as the
chimp-human and gorilla-human comparisons, in the main text, too.

The reviewer is right; we have included it in Main Text (line 120).

*Line 104: Could the pattern with the archaics be due to smaller sample size? Maybe not because the
authors try to test this for the uneven sample sizes across the great ape subspecies by subsampling the
subspecies with larger numbers, but the difference here is that there is a deep temporal separation of
these individuals, making the small samples potentially even less representative.

We agree with the reviewer.

Although we found that 5 high-diversity NHGA samples are enough to reconstruct the strong
correlations, archaic humans are expected to have lower diversity than those subspecies. Beyond the
low sample size, the archaic samples also have more fragmented sequences (1,433 Mbp of sequence
passing all filters). The archaic samples have been processed to filter out ancient DNA damage
affecting especially to C>T mutations. This filtering might be affecting mainly the CpG>T mutation
density in these samples. Thus, we cannot confidently test if the CpG>T acceleration was already
present in the human-Neanderthal ancestor or appeared after their split. Consequently, the results
obtained for archaic samples must be considered with caution and no strong conclusion, other than the
absence of strong differences, can be extracted.

Although we already mentioned the possible influence of the small sample size (Supplementary Text
line 487-488), we have now made these caveats explicit in the Main Text (lines 145-146) and this
reasoning is now expanded in Supplementary Text (lines 498-504).

*Line 182: missing period and possible missing last work (germlines)?

Thank you for finding this typo. We corrected it in Main Text (line 267).

*SI Line 130 and 145 - I think missmapping[s] should be mismapping[s] (one s)

We than the reviewer for detecting this error. Corrected in several instances across both the Main Text
(1) and Supplementary Text (3).

*SI Line 175: it seems like SNV should be plural (SNVs)

Thank you for finding this typo. Corrected in Supplementary Text (line 188).

REVIEWERS' COMMENTS:

Reviewer #1 (Remarks to the Author):

I believe all concerns have been sufficiently addressed.

Reviewer #2 (Remarks to the Author):

I would like to commend the authors on addressing the points in my review. I think they have done a good job of restructuring the manuscript to provide more clarity for the reader. They have also done a good job of addressing my minor points including adding additional detail and analyses where appropriate. I agree that engaging in simulations to test their hypothesis for the patterns they observe is outside the scope of their study.

Reviewer #3 (Remarks to the Author):

I am very impressed with the level of detail and thoroughness of the authors' responses to my concerns, including performing additional proof-of-concept analyses. I am satisfied with their response and the additional caveats and discussion added in text. I congratulate the authors on an exhaustive, novel, and interesting study.

Best,

Elaine Guevara

REVIEWERS' COMMENTS:

Reviewer #1 (Remarks to the Author):

I believe all concerns have been sufficiently addressed.

Reviewer #2 (Remarks to the Author):

I would like to commend the authors on addressing the points in my review. I think they have done a good job of restructuring the manuscript to provide more clarity for the reader. They have also done a good job of addressing my minor points including adding additional detail and analyses where appropriate. I agree that engaging in simulations to test their hypothesis for the patterns they observe is outside the scope of their study.

Reviewer #3 (Remarks to the Author):

I am very impressed with the level of detail and thoroughness of the authors' responses to my concerns, including performing additional proof-of-concept analyses. I am satisfied with their response and the additional caveats and discussion added in text. I congratulate the authors on an exhaustive, novel, and interesting study.

Best,
Elaine Guevara

We thank all reviewers for their kind words of support. We did not perform any further changes in the manuscript.